# Characterization of Maladaptive Processes in Acute, Chronic and Remission Phases of Experimental Colitis in C57BL/6 Mice

**DOI:** 10.3390/biomedicines10081903

**Published:** 2022-08-05

**Authors:** Elif Gelmez, Konrad Lehr, Olivia Kershaw, Sarah Frentzel, Ramiro Vilchez-Vargas, Ute Bank, Alexander Link, Thomas Schüler, Andreas Jeron, Dunja Bruder

**Affiliations:** 1Infection Immunology Group, Institute of Medical Microbiology and Hospital Hygiene, Otto-von-Guericke University, 39120 Magdeburg, Germany; 2Department of Gastroenterology, Hepatology and Infectious Diseases, Section of Molecular Gastroenterology and Microbiota-Associated Diseases, Otto-von-Guericke University, 39120 Magdeburg, Germany; 3Institute of Veterinary Pathology, Freie Universität Berlin, 14163 Berlin, Germany; 4Institute of Molecular and Clinical Immunology, Medical Faculty, Otto-von-Guericke University, 39120 Magdeburg, Germany; 5Immune Regulation Group, Helmholtz Centre for Infection Research, 38124 Braunschweig, Germany

**Keywords:** DSS colitis, colon, lamina propria leukocytes, colon inflammatory milieu, intestinal microbiome

## Abstract

Inflammatory bowel disease (IBD) is a chronic recurrent inflammatory disease with unknown etiology. Dextran sulfate sodium (DSS) induced colitis is a widely used mouse model in IBD research. DSS colitis involves activation of the submucosal immune system and can be used to study IBD-like disease characteristics in acute, chronic, remission and transition phases. Insight into colon inflammatory parameters is needed to understand potentially irreversible adaptations to the chronification of colitis, determining the baseline and impact of further inflammatory episodes. We performed analyses of non-invasive and invasive colitis parameters in acute, chronic and remission phases of the DSS colitis in C57BL/6 mice. Non-invasive colitis parameters poorly reflected inflammatory aspects of colitis in chronic remission phase. We found invasive inflammatory parameters, positively linked to repeated DSS-episodes, such as specific colon weight, inflamed colon area, spleen weight, absolute cell numbers of CD4+ and CD8+ T cells as well as B cells, blood IFN-γ level, colonic chemokines BLC and MDC as well as the prevalence of Turicibacter species in feces. Moreover, microbial Lactobacillus species decreased with chronification of disease. Our data point out indicative parameters of recurrent gut inflammation in context of DSS colitis.

## 1. Introduction

Inflammatory bowel diseases (IBD) include Crohn’s disease (CD) and Ulcerative Colitis (UC) and both are chronic disorders of the digestive tract, characterized by diarrhea, rectal bleeding, abdominal pain, fatigue and weight loss [1]. If not treated properly early after disease onset, IBD can cause severe complications such as intestinal fibrosis, fistulas and colon cancer [2]. IBD patients suffer from repeating cycles of disease relapse followed by remission phases of unpredictable lengths. To date, IBD is believed to be a multi-causal disease with many influential factors including genetic and immunological predisposition, dietary and microbial gut flora composition [3].

Dextran sulfate sodium (DSS) induced colitis is one of many animal models [4] that induce intestinal inflammation with many disease characteristics mimicking those of IBD patients and UC in particular. DSS colitis has been extensively used in mice since a first report in 1990 [5]. Since then, DSS has been used in numerous studies aiming to understand mechanisms of IBD pathogenesis. Accordingly, detailed knowledge on different biological aspects of DSS-induced colitis in mice is available to date, rendering the DSS model an indispensable research tool that is continuously used for specialized basic research questions as well as for preclinical testing of new IBD therapeutics and therapies [6,7,8]. DSS is appreciated for its applicative simplicity and reproducibility in establishing different stages and degrees of colitis by varying DSS concentration or frequency and duration of DSS dosing [9].

Mechanistically, DSS is toxic to the intestinal epithelial barrier, ultimately allowing translocation of luminal bacteria and antigens into the colonic mucosal immune compartment. Thereby, DSS triggers microbiota-driven, as well as immune response related, pathology.

Experimentally, acute DSS inflammatory symptoms in C57BL/6 mice are typically induced by providing between 2 and 3% (m/v) DSS in drinking water for about a week [9]. Non-invasive monitoring of the DSS colitis is commonly achieved by assessing the body weight loss, feces consistency and the amount of blood spotting, with those parameters combined into a disease activity index (DAI). Repeatedly administered DSS, under compliance of remission periods, can be used to also mimic chronic colon inflammation. In chronic DSS settings, body weight loss occurs in each DSS dosing period and a few days afterwards with mice ultimately regaining their initial body weight during remission periods with conventional water uptake [10,11].

Though body weight loss [12] and DAI are useful for evaluating the immediate symptomatic consequences of DSS consumption, it remains difficult to delineate the state of intestinal inflammation and/or systemic immune response, solely from those non-invasive disease criteria.

The immunological dimension of acute, chronic, early/late remission phases of DSS colitis is highly dynamic and only assessable via invasive elaborate analysis methods, requiring gut tissue isolation from experimental animals to, e.g., gain access to histological sections and leukocyte isolates to perform flow cytometry analysis of colon-infiltrating immune cell subsets or gut tissue homogenates to measure local inflammatory cytokine/chemokine content.

Further invasive inflammatory indicators are colon length and the specific colon weight, as the colon becomes shorter in acute colitis phases and shows increased specific colon weight [13,14,15]. DSS colitis mouse models have also been used to study the influence of the intestinal flora on the course and the extent of intestinal inflammation [16,17].

Currently, there are only a few studies that allow a simultaneous view of the previously mentioned non-invasive and invasive symptomatic, inflammatory and microbial hallmarks of experimental DSS colitis in mice, encompassing a single (acute) and multiple (chronic) DSS treatment(s) as well as their according remission phases. Knowledge regarding immunological changes taking place during transition from acute to chronic intestinal inflammation remains largely fragmentary. However, deeper insights into the pathogenesis, chronification, and remission of IBD patients is crucial for the future development of targeted therapies that allow specific intervention early on after disease onset to prevent or at least delay disease progression to chronification. Thus, we present here simultaneous systematic assessment of body weight loss, DAI, colon shortening and colon weight alongside colon histopathological scoring, flow-cytometry-based analysis of infiltrating colon immune cells, multiplex cytokine/chemokine milieus and rRNA-gene-based analysis of the gut microbiome in acute, chronic, acute remission and chronic remission phases of experimental DSS colitis in C57BL/6J mice.

## 2. Materials and Methods

Animal experiments were approved by the local authorities (Landesamt für Verbraucherschutz, Saxony-Anhalt, Germany) under license ID AZ 42502-2-1521 Uni MD.

### 2.1. Mice, Colitis Induction and Disease Scoring

All animal experiments were performed using 11 weeks-old female C57BL/6JRj (Janvier Labs) mice delivered from the same breeding barrier. Mice were kept under specific pathogen-free conditions at the central animal facility of the Medical Faculty of the Otto-von-Guericke-University, Magdeburg. Mice were sacrificed by CO_2_ inhalation. Acute colitis was induced by the application of 2% (*w*/*v*) DSS (36,000–50,000 Da, MP Biomedicals, Eschwege, Germany) into the drinking water ad libitum for 6 days. To allow recovery from acute colitis, mice were subsequently provided with DSS-free drinking water for an additional 19 days. Chronic colitis was established by in total three consecutive cycles of DSS administration (1.7% *w*/*v*) for 6 days (ad libitum access), including a 14-day interim recovery period with a supply of DSS-free drinking water after the first and second DSS cycle. To establish a remission state from chronic colitis, mice received normal drinking water for a further 21 days after completion of the final DSS cycle. Control mice received normal drinking water during the entire duration of the experiment. Relative body weight loss was monitored, and disease activity index (DAI) was determined by combining relative body weight loss, stool consistency and the extent of blood in feces. Occult blood in the feces samples was determined using the hemoCARE assay (CARE diagnostica, Voerde, Germany). DAI scoring was performed as shown elsewhere [18]. Colon and cecum length were measured with a ruler after slightly stretching the colon on a paper wipe. Colon tissue weight was determined using an analytical balance after removing the cecum and cleaning the colon from feces.

### 2.2. Histopathology of Colon

After mice were sacrificed, colons and cecums were put in histology cassettes, kept in 4% PFA and were finally embedded in paraffin. Formalin fixed paraffin embedded (FFPE) tissue slices, were stained with hematoxylin and eosin (HE). For histopathological scoring, the following criteria were used: infiltration of immune cells [19], epithelial damage [19,20], extent of damage [21], percentage of tissue having inflammation [21]. Details of scoring criteria are summarized in Table 1.

### 2.3. Isolation of Lamina Propria Leukocytes from Colon

Colonic lamina propria leukocytes were isolated according to the Lamina Propria Dissociation Kit protocol from Miltenyi Biotec (Bergisch-Gladbach, Germany). Incubation was performed in a 37 °C incubator with sample agitation. Total leukocytes were purified from the digested tissue suspension using Percoll (GE Healthcare, Uppsala, Sweden) density (~1.041 g/mL) gradient centrifugation (20 min, 1800 rpm, room temperature, without rotor brake). Supernatant was discarded, the leukocyte pellet was washed once with phosphate buffered saline (PBS). Half the volume of the isolated leukocyte suspension was used for subsequent fluorescent activated cell sorting (FACS) analysis.

### 2.4. Isolation of Splenocytes and Mesenteric Lymph Node Cells

Spleen weight was determined using an analytical balance. The spleen was placed in a 100 µm cell strainer in a petri dish with sterile PBS and crushed through the strainer using a syringe plunger. Cell suspension was collected in a 15 mL sample tube and centrifuged (10 min, 1200 rpm, 4 °C). Supernatant was discarded and for erythrocyte lysis, the cell pellet was resuspended in 5 mL 0.2% NaCl. After 15–30 s, 5 mL 1.6% NaCl was added to stop the reaction. Cells were passed through a cell strainer (70 µm) and centrifuged (10 min, 1200 rpm, 4 °C). Supernatant was removed, the pellet was resuspended in FACS buffer (PBS, 2 mM EDTA, 2% FCS) and an aliquot was stained for FACS analysis.

Mesenteric lymph nodes (MLNs) were crushed through a cell strainer (70 µm) using a syringe plunger into a 6-Well plate containing PBS. Cell suspension was collected, and the tube was centrifuged (10 min, 1200 rpm, 4 °C). Supernatant was removed, the pellet was resuspended in FACS buffer, and an aliquot was stained for FACS analysis.

### 2.5. Flow Cytometry

Immune cell subtypes were identified by staining of cell surface lineage markers with fluorochrome labeled antibodies and an Attune NxT flow cytometer (Thermo Fisher, Carlsbad, CA, USA). For antibody staining, cells were transferred into in a 96-well plate. Cells were pre-treated with TruStain FcX (anti-mouse CD16/32) antibody (BioLegend, San Diego, CA, USA), Fixable Viability Dye eFluor 506 (eBioscience, San Diego, CA, USA) or equivalent Zombie Aqua Fixable Viability dye (BioLegend, San Diego, CA, USA). For absolute cell counting of colon leukocytes, CompBead Plus Negative Control Beads (BD Biosciences, Heidelberg, Germany) were used. A suspension of CompBeads was prepared in PBS and the bead concentration was manually determined using a Neubauer hemocytometer. To each colon leukocyte sample a defined number of CompBeads was added (20,000 beads/well). TruStain FcX blocking and viability staining was incubated for 10 min at 4 °C in the dark. Cells were washed once with FACS buffer (centrifugation: 1200 rpm, 5 min, 4 °C).

Cells were stained with antibodies directed against the following surface markers (CD170-FITC (clone: S17007L), Ly6C-PerCP-Cy5.5 (clone: HK1.4), CD24-PE (clone: M1/69), CD8a-PE-Cy5 (clone: S3-6.7), CD4-PE-Cy7 (clone: RM4-5), CD11c-APC (clone: N418), Ly6G-AF700 (clone: 1A8), CD11b-APC-Cy7 (clone: M1/70), CD64-BV421 (clone: X54-5/7.1), CD45-BV605 (clone: 30-F11) and IA/IE-BV711 (clone: M5/114.15.2) diluted in FACS buffer for 10 min at 4 °C in the dark. Cells were washed once with FACS buffer (PBS, 2 mM EDTA, 2% FCS) and finally fixed with 2% paraformaldehyde (in PBS) for ~20 min at 4 °C in the dark, centrifuged (1200 rpm, 5 min, 4 °C) and analyzed on an Attune NxT flow cytometer (Thermo Fisher, Carlsbad, CA, USA). Single staining for compensation was performed using UltraComp eBeads (Invitrogen, Carlsbad, CA, USA). Single staining for the viability dye was performed on splenocytes. Flow cytometric data were analyzed using the FlowJo (version 9.9.6, Ashland, OR, USA) software. Marker choice and gating strategy are based on data published by Yu Y.R. et al. [22]. Details on the gating strategy of colon leukocytes are described in Appendix A.

Absolute cell count for colon samples was calculated as follows: (V/v) ∙ s ∙ (B/b), with V: total volume of FACS buffer used to resuspend the cell pellet, v: partial volume used for FACS staining, s: events obtained for a given immune cell population, B: added CompBeads number (20,000 beads/well), b: acquired effective CompBeads number. In case an immune cell population was counted with less than 100 events per sample, this population was excluded from calculations of absolute cell number calculation and frequency of CD45+ cells. In case a sample had less than 2000 events of CD45-positive cells, the sample was completely discarded from the calculation of absolute cell numbers and frequency.

### 2.6. Quantification of Cytokine and Chemokine Levels in Plasma and Colon

Blood samples were collected by heart puncture immediately after euthanization of mice and were transferred into Eppendorf tubes containing 20 µL EDTA (0.5 M). To obtain platelet-free plasma, samples were centrifuged for 10 min at 500× *g* at 4 °C, supernatants were transferred into new tubes and centrifuged once more at maximum speed at 4 °C. Plasma samples were stored at −80 °C.

Colons were excised, feces was removed and colons were washed in PBS, cut open longitudinally and cut into small pieces. Tissue pieces were mingled and split into four approximately equal portions. One fraction of the tissue was transferred into Lysis Matrix Tubes D (MP Biomedical, Eschwege, Germany) containing 500 µL NP40 Cell lysis buffer (Thermo Fischer, Carlsbad, CA, USA) with protease inhibitors (cOmplete Mini Protease Inhibitor Tablets, Roche, Basel, Switzerland; one tablet for 7 mL lysis buffer). Tissue was homogenized using a FastPrep instrument (MP Biomedical, Eschwege, Germany) for 40 s at a speed of 6.0 m/s. Homogenized samples were centrifuged at 10,000× *g* for 10 min and supernatant was collected and stored at −80 °C. Remaining colon tissue portions were kept frozen at −80 °C as backup material.

To standardize subsequent cytokine/chemokine detection in tissue samples, total protein concentration was determined using BCA assay (see below). To determine cytokine and chemokine concentrations in blood plasma and tissue homogenates, LEGENDplex mouse inflammation panel (IL-23, IL-1α, IFN-γ, TNF-α, MCP-1, IL-12p70, IL-1β, IL-10, IL-6, IL-27, IL-17A, IFN-b, GM-CSF) and LEGENDplex Mouse Proinflammatory Chemokine Panel (RANTES, MCP-1, IP-10, Eotaxin, TARC, MIP-1α, MIP-1β, MIG, MIP-3α, LIX, KC, BLC, MDC) (both BioLegend, San Diego, CA, USA) were used following the manufacturer’s recommendations. Analytes were quantified using an Attune NxT flow cytometer (Thermo Fisher, Carlsbad, CA, USA). Data were analyzed using BioLegend’s cloud bases LEGENDplex analysis software (version 2021.07.01). Cytokine and chemokine levels were normalized to total protein concentration.

### 2.7. BCA Assay

Total protein concentration of tissue homogenates was determined by Pierce BCA Protein Assay Kit (Thermo Fisher, Carlsbad, CA, USA). Pierce Bovine Serum Albumin Standard Ampules (Thermo Fisher, Carlsbad, CA, USA) was used as standard. Standard and samples were diluted by isotonic saline 0.9% (Fresenius Kabi GmbH, Bad Homburg, Germany).

### 2.8. Microbiota Analysis

Feces samples were collected at indicated time points and stored at −80 °C until DNA extractions. DNA was extracted using QIAamp Fast DNA Stool Mini Kit (Qiagen, Hilden, Germany). Amplicon libraries were generated by amplifying the V1-V2 region of the 16S rRNA gene after 20 cycles PCR reaction using the 27F and 338R primers and sequencing on an Illumina MiSeq (2 × 250 bp, Illumina, Hayward, CA, USA) [23,24].

All FastQ files, generated after sequencing and demultiplexing, were analyzed using dada2 package version 1.14.1 in R [25], resulting in a single table containing all samples with the sequence reads (Phylotype) and their abundance. Sixty-nine samples were resampled to equal the smallest library size of 11,414 reads using the phyloseq package version 1.30.0 in R [26]. Sequence reads were taxonomically annotated with the ribosomal database project [27], based on the naïve Bayesian classification [28] with a pseudo-bootstrap threshold of 80%. In addition, the first 100 more abundant phylotypes in the cohort were manually annotated with the NCBI database using the Blast-Algorithm [29,30]. Microbial communities were analyzed at the taxonomic rank of genus in relative abundances (expressed as percentages, Appendix A).

Principle component (PCO) clustering and multivariate tests (Anosim and Permanova) were performed in Primer 7, based on a Bray–Curtis resemblance measurement at the taxonomic rank of genus [31,32,33]. Differences in the distribution of genera between consecutive experimental days of each mouse group were calculated by the Mann–Whitney U unpaired test with 95% confidence interval, by using the “ExactRankTest” package version 0.8–29 in R. The resulting *p* values were corrected by applying the Benjamini–Hochberg false-discovery rate correction (desired FDR = 5%) [34].

### 2.9. Statistics

If not otherwise indicated, visualization and statistical analyses were performed by GraphPad Prism Software version 8 or 9.2 (GraphPad Software, Inc., La Jolla, CA, USA). *p* < 0.05 was considered statistically significant.

## 3. Results

To establish acute and chronic colitis as well as the respective remission stages, mice received a single treatment (“acute”) or three consecutive treatments (“chronic”) with DSS drinking water (acute: 2%, chronic: 1.7%) for six days. Mice in the chronic cohort had two weeks of remission time in between the DSS periods and mice in the “chronic remission” cohort were given a final remission period of about 3 weeks, as outlined in the treatment chart in Figure 1A. Mice in the “acute remission” cohort experienced a single DSS period followed by a remission phase of about 3 weeks on normal drinking water. Considering durations of the acute and chronic experimental timelines, age-matched healthy control groups were used, respectively.

### 3.1. DSS-Induced Colitis Is Associated with Sustained Intestinal and Systemic Inflammation Even after Normalization of Outwardly Visible Disease Symptoms

Initially, disease progression upon DSS treatment and remission periods was monitored by determining acknowledged non-invasive parameters such as relative body weight loss and disease activity index (DAI), which evaluates feces consistency and blood spotting. During periods of DSS administration, mice lost on average about 10% of their body weight (Figure 1B,D), accompanied by increased DAI (Figure 1C,E), but they largely recovered their original body weight after up to 7 days post DSS treatment. Maximal body weight loss typically occurred up to four days after the mice were given normal drinking water again (Figure 1D). This pattern was observable after all three DSS periods. In this regard, it is worth mentioning that the maximal DAI value was correlated with the last day of DSS dosage of a given DSS period and thus typically preceded the day of maximal body weight loss, indicating that both parameters are temporally out of phase (Figure 1D,E). Although mice that experienced three DSS periods, including the final remission period, eventually were able to more than compensate interim body weight loss, their feces-related disease activity index remained slightly elevated, indicative of low residual colon inflammation (Figure 1D,E).

The invasively accessible parameter of colon shortening is a good indicator of intestinal inflammation and conversely the normalization of colon length indicates post-inflammatory tissue recovery. Accordingly, we found significant colon shortening of about 2 cm in acute and chronic DSS conditions (Figure 1F). Colons in the remission phase of acute DSS almost turned back to their initial lengths, but colons from mice in the chronic remission phase were still somewhat shorter than colons of both untreated control groups (Figure 1F), which is in line with the previously mentioned residual DAI score at this point in time.

Colon inflammation is associated with the accumulation of lamina propria leukocytes and the subsequent increase of specific tissue weight of cleaned feces-free colons. Here we found even stronger evidence of residual colon inflammation, contradicting a presumed recovery based on just the body weight. Mice from chronic but especially both remission conditions exhibited significantly denser colons compared to both control groups (Figure 1G). Fittingly, 3 weeks after ending the third DSS cycle and its remission phase, mice still exhibited significantly increased spleen weights (Figure 1H), indicative of persistent low-level systemic inflammation.

On microscopic inspection of the colons by histology and systematic histopathological scoring, we found that even after the 3 week recovery period, in both the acute and chronic remission group, the histopathology score was still increased (Figure 1I,J). More meaningful, however, for evaluating the chronic remission condition was the relative colon area that is affected by inflammation (Figure 1K). Here, the affected area became significantly larger after the third DSS cycle and remained similarly large in the according remission phase 3 weeks later (Figure 1K). Colon inflammation at this point in time is a conceivable driver of the previously mentioned low level systemic inflammation.

Together, our data demonstrate that the inflammatory aspect of recovery from acute and chronic DSS-induced colitis cannot be delineated solely from non-invasive scoring of body weight and DAI.

### 3.2. The Composition of Lamina Propria Leukocytes Changes in the Course of DSS-Induced Colitis

As we observed increased immune cell recruitment implied by specific colon weights (Figure 1G) and colon histology (Figure 1I,J), we set out to determine identities of colonic immune cells at the different acute, chronic and remission stages of DSS-induced colitis.

Stages of colitis were induced according to Figure 1A and total colonic lamina propria leukocytes were isolated from enzymatically digested colons at indicated time points. Subsequently, we performed multicolor FACS profiling of lineage (lin) cell markers to identify nine major canonical immune cell subsets (B cells, lin- lymphocytes, macrophages, CD4+ and CD8+ T cells, eosinophils, dendritic cells, monocytes and neutrophils). Similar event counts from analyzed colon samples per condition were merged in silico and lineage marker expression was used for t-SNE embedding of cell subsets. This qualitative visual overview showed remarkable changes in the cell composition at different disease stages that are, at least in part, sustained even during the recovery phase of chronic DSS colitis (Figure 2A).

Among the most obvious changes in leukocyte composition observed in the t-SNE plots were the neutrophils. They were not detectable in both control groups, but their proportion increased during acute and remained elevated during chronic intestinal inflammation. Of note, neutrophilia was still evident three weeks after termination of DSS treatment in both the acute and chronic remission conditions (Figure 2A).

Among the lymphocytes, CD4+ T cells particularly increased in the recovery stage after acute DSS colitis and stayed high during chronic disease stages. To our surprise, the proportion of eosinophils dramatically expanded during the acute phase of DSS colitis, but not in any of the other analyzed disease stages (Figure 2A).

t-SNE-based analyses were complemented by analysis of mouse-individual leukocyte frequencies (Figure 2B and Appendix A) and quantification of absolute cell numbers (Figure 2C and Appendix A). As already observed in the t-SNE plots, neutrophils were recruited to the colon early after onset of DSS-induced inflammation (day 6) and remained clearly increased in frequency and absolute cell number in all other disease stages (Figure 2B,C). For other myeloid cell populations, t-SNE analysis was also in line with quantitative data of cellular subsets. Here, a significant increase was found for eosinophils in the acute DSS, while monocytes were increased both in the acute and chronic stages of DSS colitis but returned to normal levels during the recovery phases (Figure 2B,C). Frequencies and total cell numbers of adaptive lymphocyte subsets such as CD4+, CD8+ T cells and B cells increased especially in the chronic phase and remained increased even after the 3-week recovery phase (Figure 2B,C). Despite distinct alterations in their relative frequencies, no clear changes in absolute cell numbers of double negative lymphoid cells, dendritic cells and macrophages were observed.

Next to local changes in cellularity within the colonic tissue itself, its draining mesenteric lymph nodes (MLN) similarly show elevated frequencies of B cells in the acute remission and chronic stages. Moreover, frequencies of lymph node DCs are elevated in the chronic and chronic remission phase of DSS treatment (Appendix A).

Immune cell composition of the spleen, representing the systemic inflammatory response, also shows perpetuating adaptations to repeated DSS episodes. Here, e.g., the neutrophil frequencies are particularly elevated in the chronic DSS phase (Appendix A).

Taken together, chronic adaptation of the lamina propria immune cell composition clearly involves cells of the adaptive immune system, signified by enhanced pools of CD4+ and CD8+ T cells as well as B cells, with all three lymphocyte subsets strongly expanding only after the acute remission phase.

### 3.3. Acute and Remission Stages of DSS-Induced Colitis Are Associated with Distinct Local Cytokine/Chemokine Milieus

Since we observed sustainably altered cell composition in the colon and systemic inflammatory changes in the spleen, we next analyzed whether potential changes in the local cytokine and chemokine profile during different disease stages might correlate with observed cellular tissue-influx patterns. To this end, selected inflammation-related cytokines and chemokines were quantified in supernatants of colon homogenates using a bead-based multiplex assay. Indeed, the local inflammatory colon environment shows some common protein expression patterns regarding a significant increase in the production of the chemokines and cytokines under consideration. The most frequently occurring pattern is tightly correlated with the administration of DSS in the drinking water. Here, the chemokines Rantes, KC, MCP-1, MIG, IP-10 and MIP-1α as well as the cytokines IL-1α, IFN-γ, IL-6 and TNF-α each showing significantly increased protein tissue expression at the end of single and triple DSS episodes (Figure 3A,B and Appendix A). The frequency of DSS administration does not seem to have an influence on the previously mentioned proteins, as no statistically significant differences were observed between acute (one DSS cycle) and chronic DSS colitis (three DSS cycles).

In contrast, systemic IFN-γ levels in the plasma are clearly elevated only in chronic and chronic remission phases, indicating that chronification of colitis by three DSS cycles enhances the degree of systemic inflammation opposing our findings in the local colonic tissue (Appendix A).

Another occurring cytokine/chemokine expression pattern relates to the two chemokines MIP-3α and BLC as well as the two cytokines IL-1β and IL-17A, which show significantly increased tissue expression only after three DSS cycles, suggesting a specific local adaptation of the colonic milieu to repeated episodes of inflammation (Figure 3A,B).

From this point of view, the expression of the chemokine MDC is particularly interesting, as it shows an increased expression especially in the chronic DSS phase as well as three weeks afterwards in the late recovery phase (Figure 3A). Of note, IL-17A similarly was slightly elevated in the chronic remission phase. IL-17A was systemically detectable also in blood plasma but did not show any significant differences between the DSS conditions (Appendix A).

Eotaxin also shows a special expression pattern among the chemokines considered, as it is significantly increased exclusively after a single DSS administration and returns to the initial level in the subsequent first remission phase (Figure 3A). As Eotaxin is a potent attractor of eosinophils, this is well in line with the previously observed infiltration of eosinophils during acute DSS colitis (Figure 2).

Taken together, residual local colon inflammation in the chronic remission phase as observed previously by other parameters can be characterized based on cytokine/chemokine expression by MDC and IL-17A with further yet non-significant examples such as MIP-1α, IL-1α, KC, MIP-3α, MCP-1, MIG, IP-10, IL-1β and TNF-α.

However, IFN-γ is a good indicator of systemic and indirectly local colonic inflammatory adaptation as its concentration only increases in the blood plasma after three DSS periods as well as in the chronic remission phase.

### 3.4. Chronification of DSS-Induced Colitis Is Associated with the Perpetuation of Intestinal Microbiota Dysbiosis

It has been shown before that acute DSS colitis is associated with reduced diversity of bacterial species as well as marked changes in the intestinal microbiota composition [35]. However, the sustainability of intestinal dysbiosis and/or its potential gradual consolidation during colitis chronification are only poorly understood to date. To experimentally address these issues, we performed time-resolved rRNA-gene-based microbiota analysis of feces from the same mouse individuals at consecutive stages of the disease model—after the first cycle of DSS treatment (day 6), after the second (day 26) and after the third (day 47) DSS cycle and the respective recovery stages (day 20, 40 and 67, respectively).

The analysis of microbiome data obtained from this cohort showed that the intestinal microbial community consists of 89 different genera and 20 taxa not classified to genus level (Appendix A).

Principle component (PCO) cluster analysis revealed that DSS treatment and the resulting colitis clearly affects the intestinal microbiome in a cumulative fashion. In the PCO-plot (black symbols: DSS dosage, gray symbols: after DSS remission phase, red symbols: water control) this becomes evident by the fact that only microbiomes from feces of mice before receiving DSS and microbiomes of mice that are in the first remission phase (day 20, gray triangles) cluster together with the microbiomes of drinking water control mice (Figure 4A). Microbiomes of the two later (day 40 and 56) remission phases (gray plus symbols and gray squares) remain, however, positioned in between microbiomes of control and DSS consuming mice (day 6, 26 and 47) in the PCO plot (Figure 4A), thereby indicating overall perpetual microbial inflammatory adaptation, which takes hold from the second DSS episode on.

Evolving maladaptation of the microbial community with an increasing number of DSS treatments and chronicity of colitis was further supported by multivariate Anosim and Permanova tests, which statistically confirmed, firstly, microbial adaptation during each DSS dosing period (compared with water-only controls) and, secondly, impairment of the microbiome’s ability to restore the initial composition of the gut flora in the last remission phase (day 67) after three DSS cycles (Figure 4B). In more detail, this analysis revealed that while classified intestinal microbial genera in control animals remained largely stable (except around day 20, Figure 4B,C upper panel), significant differences were detectable in the DSS cohort at any time point analyzed. In general, on the genus level, the most significant microbial adaptations occurred at the end of DSS dosing periods, e.g., accompanied by a reduction of Lactobacillus and outgrowth of Prevotella species (Figure 4C). In principle, microbiome composition normalized to some extent during a 2-week interim remission period. However, the ability of the microbiome to return to the initial gut flora composition during remission phases decreased with each of the three DSS periods applied, as evidenced, for example, by a persistently reduced proportion of Lactobacilli after the last remission phase on day 67 (Figure 4C).

Amongst classified bacterial genera, the most dynamic DSS-induced changes in the microbiome composition attributed to Lactobacillus, Prevotella, Limosilactobacillus and Turicibacter (Figure 4C,D). The relative abundance of Lactobacillus and Limosilactobacillus followed a pattern of highest abundance in healthy mice and in the recovery phases, respectively, with a failure in re-establishing initial abundancies after the third DSS cycle (Figure 4D). Abundance of the genus Prevotella followed the same course as the disease activity index (Figure 1E), with high abundance in stages of DSS-dosage and a decline back to normal in the recovery phases (Figure 4D). The Turicibacter genus was largely undetectable in healthy animals but, strikingly, its abundance in the active stages of colitis steadily increased with each DSS cycle applied but normalized during the recovery phases (Figure 4D).

Together, intestinal dysbiosis is clearly initialized by active periods of DSS-dosage in which each period has a cumulative progressive impact on microbiota composition, leading to dysbiosis in the chronic remission phase.

## 4. Discussion

From the immunological point of view, in DSS colitis studies there is dissenting usage of terminology and meaning of the concepts of “acute” and “chronic” inflammation as well as “remission phase”. Melgar and Hall et al. defined the stages of DSS colitis in C57BL/6 mice in a model with six days of DSS dosage, with respect to immune cell presence. They defined an early acute stage of the disease as ranging from day 1 to 8, based on neutrophil influx into the colon [10,36]. Moreover, they defined a late stage of acute DSS colitis from day 12 on, based on neutrophil decline and increased numbers of adaptive immune cells. From day 25 on, the chronic stage of DSS colitis is reached, encompassing significant numbers of B and T cells in colonic tissue [10,36].

However, we believe that a single DSS treatment episode is inadequate for representing the full extent of recurrent inflammatory episodes that occur in IBD patients and therefore use the term “chronic inflammation” here exclusively in the sense of cyclically repeated episodes of inflammation and that of “acute inflammation” for a single, time-limited DSS episode. Similarly, the term “remission phase” is not used in a standardized way in studies involving the DSS colitis model and is not linked to clear criteria. We use it here generally for periods following DSS dosage, although a reduction of most colitis symptoms does not occur immediately after the end of a DSS period. In this respect and in the context of our DSS colitis model, we understand “remission” here in the sense of an ongoing homeostatic process with still ongoing but declining inflammatory signatures and less as an achieved terminal state.

Non-invasive inflammatory parameters, such as weight loss and DAI, show similar trajectories regardless of the number of DSS periods undergone. Differences in this aspect only become apparent when invasively accessible inflammatory parameters such as specific colon weight, inflammatory affected histological colon area, and spleen weight are considered (Figure 1G,H,K). In the chronic remission phase, all three parameters are comparatively elevated after three cycles of DSS compared with the remission phase after a single acute DSS period. This proves that non-invasive parameters, which are common in DSS mouse models, are unsuitable to correctly predict maladaptive developments that are only established upon chronification of disease.

Most but not all of the invasively assessable inflammatory parameters of the colon analyzed in this study, e.g., immune cell counts and cytokine/chemokine milieu show predictable and stable responses during acute and chronic stages followed by a steady decline in the according remission phase, thus similarly not indicating adaptations of colon inflammation to repeated DSS cycles. Inflammatory parameters that indeed may be used as indicators of adaptation to repeated DSS-cycles in C57BL/6 mice will be discussed in the following.

As expected, cells of the adaptive immune response such as CD4+, CD8+ T cells and B cells are naturally linked to the evolving colonic mucosal immune response upon repeated cycles of DSS dosage. Our data clearly show DSS-cycle count-dependent increase of absolute cell counts of CD4+ and CD8+ T cells as well as B cells, located within the colonic lamina propria (Figure 2C). This is due to their repeated exposure to microbial antigens, leading to initial expansion and re-activation of specific T and B cell clones [37,38]. Although the establishment of acute DSS colitis does not require the presence of T or B cells in the first place as the DSS colitis model is applicable in SCID mice, lacking these cell types [39], T cells do, however, play a role in chronic DSS models with clear contributions of mixed Th1/Th2 cell responses [21]. Moreover, CD11b+ IgA-secreting B cells were reported to ameliorate DSS colitis in mice [40]. However, when experimentally restricted to the analysis of relative frequencies, e.g., from mouse colon tissue biopsies, only CD4+ T cell frequencies of around 20% (in reference to total CD45+ lamina propria leukocytes) seem to be a clear indicator of repeated DSS episodes in C57BL/6 mice (Figure 2B).

Neutrophils become a permanent colon immune cell subset from the first DSS cycle on and are present even throughout remission phases. Colonic neutrophils are important for sufficient immunothrombosis and thereby help to reduce rectal bleeding [41]. Interestingly, we also found elevated neutrophil frequencies in the spleen, especially in the chronic DSS condition (>5% of CD45+ leukocytes) (Appendix A). Similar observations were made in blood and colon biopsies of IBD patients [42]. CD177+ Neutrophils were reported to be a beneficial phenotypic adaptation in IBD colitis, with CD177-deficient mice developing more severe DSS colitis [42,43]. Since we did not check the expression of CD177 on neutrophils, it remains unclear whether the frequency of CD177+ neutrophils would be an even more specific indicator of chronic DSS effects in our DSS model that would be potentially assessable also by blood sampling of DSS treated mice.

Chronic gut inflammation by no means triggers only a locally restricted immune response. Patients with Crohn’s disease but not UC have significantly elevated levels of serum IFN-γ [44]. Similarly, we found increased protein levels of IFN-γ in blood plasma samples of mice from the late stages of our DSS model, the chronic and chronic remission conditions (Appendix A). IFN-γ is causatively linked to DSS colitis as in a model of acute DSS colitis, IFN-γ^-/-^ mice are largely protected [45]. As we found notable colonic IFN-γ expression only in active periods of DSS dosage, the plasma IFN-γ level rather than the local colon IFN-γ level may be a good inflammatory indicator for an adaptation to, or the consequence of, repeated DSS episodes.

Amongst the soluble mediators produced within colon tissue we found the chemokines BLC (also known as Cxcl13) and MDC (also known as Ccl22) to have a meaningful expression pattern, with the latter especially prominent in the chronic remission phase.

B lymphocyte chemoattractant (BLC) is already in focus in basic and clinical IBD research. In a mouse model with BLC overexpression in intestinal epithelial cells during inflammatory conditions, influx of B cells, lymphoid tissue inducer cells and NK cells with immunomodulatory and reparative functions was described [46]. Similarly, enhanced expression of the human homologue of BLC, BCA-1 was reported in colon specimens from UC patients [47]. The fact that we found BLC expression in colonic tissue to be particularly high only after three DSS cycles (Figure 3A) renders this chemokine a good indicator of inflammatory adaptation in our DSS model and moreover is a mechanistic link to the high numbers of B cells at the same point in time.

We found particularly high and stable expression of Macrophage-derived chemokine (MDC, or Ccl22) in the chronic and chronic remission condition (Figure 3A), which is a sensible criterion for a DSS-cycle-dependent inflammatory indicator. In mice, CD11c+ dendritic cells require T cell-derived GM-CSF to become the main producers of MDC in secondary lymphatic organs during homeostasis [48]. Under inflammatory conditions, MDC is a potent T cell chemoattractant also produced by colon epithelial cells, e.g., in response to pro-inflammatory cytokines or infection with entero-invasive bacteria [49]. Recently, the Ccl22/Ccr4 axis was identified as an immune checkpoint controlling intestinal T cell and Treg cell immunity with Ccl22-deficient mice being more susceptible to DSS inflammation [50]. Hence, MDC expression in colonic tissue provides a mechanistic link to the matching, elevated lamina propria T cell counts in our chronic and chronic remission DSS model.

Our analysis of the microbial composition allows further insights into consequences of repeated DSS cycles for the gut commensals. Beneficial Lactobacilli and Limosilactobacilli steadily reduced with each further DSS cycle and were not able to regain their initial level. In turn, harmful species belonging to the Turicibacter genus continuously expanded in frequency in each DSS consumption period (Figure 4D). Thus, the prevalence of Lactobacilli and Turicibacter species are easy-to-access feces-indicators of colon adaptation to consecutive inflammatory episodes. Similar results have been reported previously [51] and further studies have been successfully undertaken to, e.g., complement the DSS-shaped microbiome with probiotic Lactobacillus acidophilus XY27 [52] or Limosilactobacillus reuteri EFEL6901, resulting in decreased expression of pro-inflammatory cytokines TNF-α and IL-1β, and higher levels of anti-inflammatory IL-10 in the colon [53]. Furthermore, Lactobacillus species are related to many other beneficial host/microbiome interactions taking place at the epithelial barrier including, e.g., the enhancement of mucus production, the enhanced release of anti-microbial peptides and a higher prevalence of luminal secretory immunoglobulin A, as intensively reviewed by Dempsey and Corr [54]. However, little is known about actual microbial mechanisms leading to the perpetuating decrease of Lactobacilli during DSS colitis. Still, it is conceivable that many of the known beneficial contributions of this bacterial genus might be critically missing within the inflamed colon and thus justify further elaborate research in this regard.

## 5. Conclusions

In conclusion, our results point out indicators of disease progression in experimental ulcerative colon inflammation and provide clues to major immunological characteristics in different stages of the DSS model and their transition. We show that colonic maladaptation requires two or more DSS periods. Thus, our study may help to improve experimental IBD animal models to enhance their clinical relevance, as well as to identify novel therapeutic targets to prevent disease progression and chronification of colitis in IBD patients.

## Figures and Tables

**Figure 1 biomedicines-10-01903-f001:**
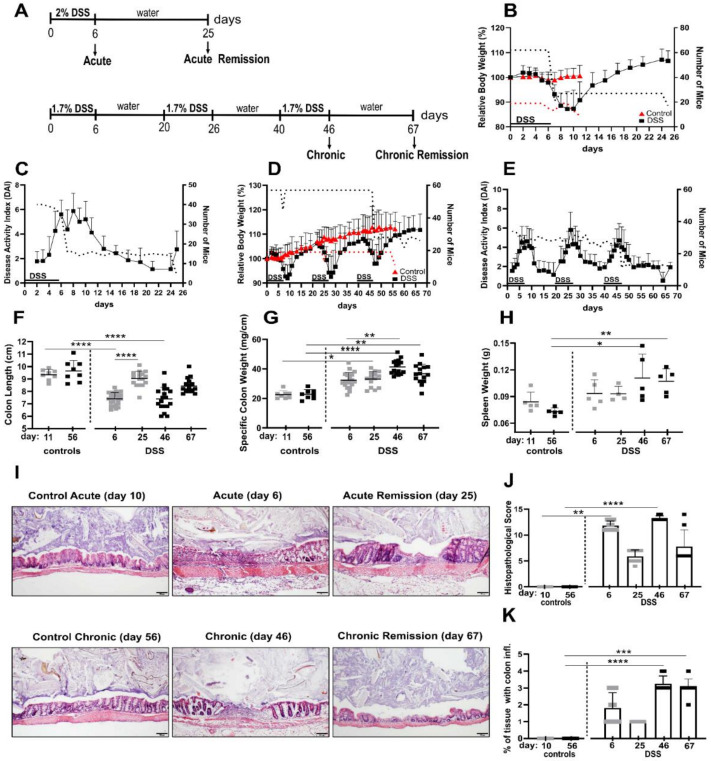
Evaluation of disease activity and histopathological alterations at different stages of dextran sulfate sodium (DSS) colitis. (**A**) Experimental design to induce consecutive stages of DSS colitis. (**B**,**C**) Relative body weight (*n* = 17–62 mice, pooled from 3–5 independent experiments) and disease activity index (DAI; *n* = 4–40 mice from 1–3 independent experiments) of mice after a single treatment with 2% DSS water for 6 days (acute and acute remission) and relative body weight (*n* = 9–19 mice pooled from 2–4 independent experiments) of mice for control acute group. (**D**,**E**) Relative body weight (*n* = 22–57 mice from 3–4 independent experiments) and DAI (*n* = 11–34 mice from 1–2 independent experiments) of mice that received three consecutive cycles of 1.7% DSS water (chronic and chronic remission) and relative body weight (*n* = 11–19 mice pooled from 2–4 independent experiment) of mice for control chronic group. (**F**–**H**) Colon length, specific colon weight and spleen weight for all mouse groups. Colon length was obtained from 2 independent experiments (*n* = 8–18 mice) except for the acute DSS colitis group which contains data from 3 independent experiments (*n* = 24 mice). Specific colon weight was obtained from two independent experiments (*n* = 8–16 mice) except for the acute DSS group which contains data from 3 independent experiments (*n* = 20 mice). Spleen weight was obtained from one experiment (*n* = 4–5 mice). (**I**) Representative histological pictures of the colons taken after hematoxylin and eosin staining (scale bar: 100 µm). (**J**,**K**) Histopathological score and percent of tissue exhibiting colon inflammation during different stages of colitis. Histology results were obtained from 2 independent experiments for all mouse groups (*n* = 8 mice) except for the acute DSS colitis group which contains data from 3 independent experiments (*n* = 10 mice). Data for control groups are from one experiment (*n* = 6). (**B**–**E**) Dotted lines represent the number of mice for each stated time point. (**F**–**H**,**J**,**K**) Grey squares represent control acute, acute and acute remission conditions and black squares represent control chronic, chronic and chronic remission groups. Data represent mean with error bars indicating standard deviation. Significance was calculated by Kruskal–Wallis test. All groups were compared with each other, significance is indicated only for relevant groups. **** *p* < 0.0001, *** *p* < 0.001, ** *p* < 0.01, * *p* < 0.05.

**Figure 2 biomedicines-10-01903-f002:**
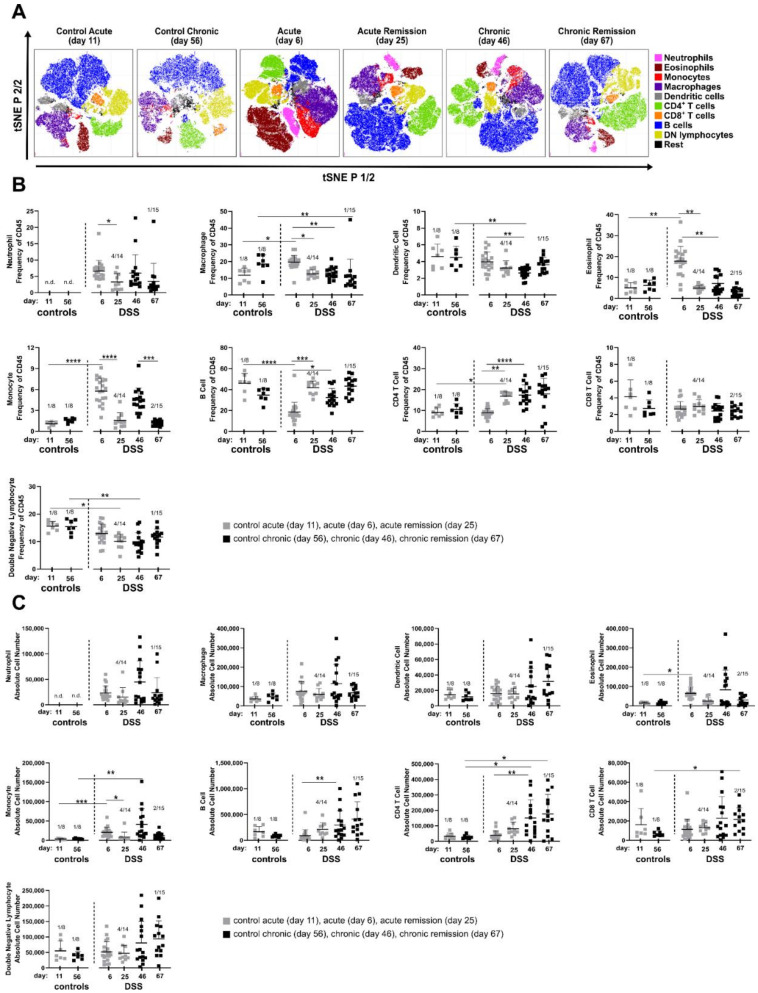
Characterization of innate and adaptive immune cell composition in the colon during consecutive stages of dextran sulfate sodium (DSS) colitis. DSS colitis and respective remission stages were induced as indicated in Figure 1A. At indicated time points, mice were sacrificed and colonic leukocytes were isolated followed by antibody staining and flow cytometry analysis. Unsupervised clustering of immune cell subsets was performed by t-distributed stochastic neighbor embedding (t-SNE) and cell subsets were identified and color-coded based on manual gating. (**A**) Representative t-SNE plots. (**B**) Frequency of indicated immune cell subsets within the CD45+ cell pool. (**C**) Absolute numbers of the indicated immune cell types. Data were obtained from 2 independent experiments (*n* = 8–16 mice), except for acute DSS colitis for which data were obtained from 3 independent experiments (*n* = 20 mice). Data represent mean with error bars indicating standard deviation. n.d.: not detectable. Number of excluded samples is indicated on top of each condition (number of excluded sample/total sample number). Significance was calculated by Kruskal –Wallis test. All groups were compared with each other, significance is indicated only for relevant groups. **** *p* < 0.0001, *** *p* < 0.001, ** *p* < 0.01, * *p* < 0.05.

**Figure 3 biomedicines-10-01903-f003:**
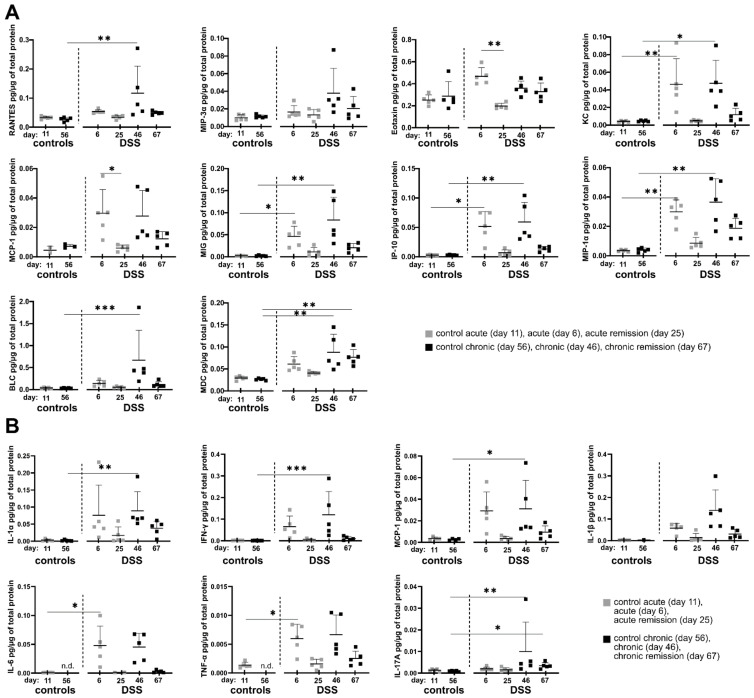
Quantification of chemokines and cytokines in colon tissue during consecutive stages of colitis. Dextran sulfate sodium (DSS) colitis and respective remission stages were induced as indicated in Figure 1A. At indicated time points, mice were sacrificed, and chemokine and cytokine levels were determined in colon homogenates and normalized to total protein amount. Depicted data match the following criteria: cytokine/chemokine concentration is above the limit of quantification (LOQ) in at least 4 out of 5 samples (80 %) in at least one condition. Samples with final concentrations that were under the limit of detection (LOD) in at least one out of two technical replicates were excluded. (**A**) Chemokine levels (*n* = 5 per group). (**B**) Cytokine levels (*n* = 5 per group). Data represent mean with error bars indicating standard deviation. n.d.: not detectable. Significance was calculated by Kruskal–Wallis test. All groups were compared with each other, significance is indicated only for relevant groups. *** *p* < 0.001, ** *p* < 0.01, * *p* < 0.05.

**Figure 4 biomedicines-10-01903-f004:**
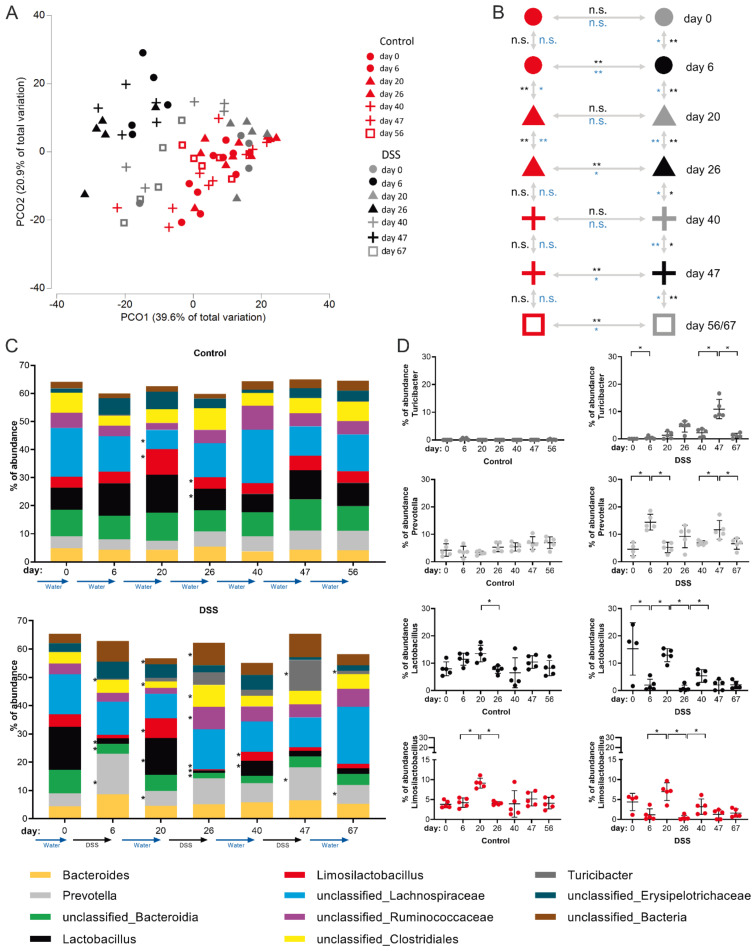
Microbiota analysis in feces samples from different stages of dextran sulfate sodium (DSS) induced colitis and control groups. DSS colitis was induced as indicated in Figure 1A (lower timeline). Control mice received normal drinking water. At indicated time points, feces were collected and subjected to Illumina MiSeq sequencing. (**A**) Principle component clustering of all feces samples. (**B**) Results of multivariate Anosim (black) and Permanova (blue) tests between the control group (normal drinking water) and the different DSS groups for each experimental day, based on a Bray–Curtis-resemblance measurement at the taxonomic rank of genus. n.s.: not significant. (**C**) Relative abundance of the global bacterial community at the taxonomic rank genus and (**D**) relative abundance of selected genera for each mouse. Data are from *n* = 5 mice per group. Statistical significance indicated by * *p* < 0.05 and ** *p* < 0.01.

**Table 1 biomedicines-10-01903-t001:** Histological scoring.

Histological Changes	Score: 0	Score: 1	Score: 2	Score: 3	Score: 4
**Infiltration of** **immune cells**	no inflammation	around crypt base	into mucosa	extensive mucosal infiltration and oedema	into submucosa
**Epithelial damage and** **loss of goblet cells**	Intact	slight loss of goblet cells	considerable loss of goblet cells and slight loss of intestinal crypts	extensive loss of intestinal crypts	
**Extent**	None	mucosa	mucosa and submucosa	Transmural	
**Percent involvement**		1–25%	26–50%	51–75%	76–100%

## Data Availability

Data is contained within the article and Appendix A. Further data are available on request from the corresponding author.

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
