# Peer review of "Characterization of Maladaptive Processes in Acute, Chronic and Remission Phases of Experimental Colitis in C57BL/6 Mice"

_biomedicines, 2022, doi:10.3390/biomedicines10081903_

Round 1

Reviewer 1 Report

The manuscript authored by Elif Gelmez and colleagues investigates both, non-invasive and invasive parameters that reflects inflammatory aspects in acute, chronic and remission stages of colitis by employing DSS-induced colitis model in mice.

Authors show that non-invasive parameters (such as body weight and DAI) cannot reliably predict maladaptive responses upon chronification process of DSS-induced colitis. In terms of invasive parameters, authors have identified specific changes at the acute, chronic and remission stages of DSS-induced colitis in (1) the composition of lamina propria immune cell lineages, (2) production of cytokines/chemokines in the colon, spleen, and blood plasma as well as (3) the intestinal microbiota.

Overall, the experimental plan is logical, experiments are well described, and the conclusions are reasonable based on the results presented. The manuscript provides interesting insights into the DSS-induced (acute/chronic) colitis model in mice and for the Gastrointestinal field.

I would suggest the authors to add summary tables to readily appreciate the changes observed for leukocyte composition and chemokines/cytokines during the consecutive stages of DSS -induced chronic colitis and remission.

Author Response

Point-to-point reply - Reviewer 1:

Reviewer: The manuscript authored by Elif Gelmez and colleagues investigates both, non-invasive and invasive parameters that reflects inflammatory aspects in acute, chronic and remission stages of colitis by employing DSS-induced colitis model in mice. Authors show that non-invasive parameters (such as body weight and DAI) cannot reliably predict maladaptive responses upon chronification process of DSS-induced colitis. In terms of invasive parameters, authors have identified specific changes at the acute, chronic and remission stages of DSS-induced colitis in (1) the composition of lamina propria immune cell lineages, (2) production of cytokines/chemokines in the colon, spleen, and blood plasma as well as (3) the intestinal microbiota. Overall, the experimental plan is logical, experiments are well described, and the conclusions are reasonable based on the results presented. The manuscript provides interesting insights into the DSS-induced (acute/chronic) colitis model in mice and for the Gastrointestinal field.

Authors: We would like to thank the reviewer for the positive feedback on our work and hope to sufficiently implement all recommendations in the following.

Reviewer: I would suggest the authors to add summary tables to readily appreciate the changes observed for leukocyte composition and chemokines/cytokines during the consecutive stages of DSS - induced chronic colitis and remission.

Authors: We thank the reviewer for the suggestion to provide tables summarizing data from leukocyte and chemokine/cytokine analyses shown in Figures 2 and 3. Accordingly, we provided supplementary tables 2 and 3 summarizing significant differences in reference to the control groups. We referenced the new supplementary tables in the main document at appropriate text positions and modified the “Supplementary Materials” statement in the main document accordingly.

Reviewer 2 Report

Manuscript title: Characterization of Maladaptive Processes in Acute, Chronic and Remission Phases of Experimental Colitis in c57bl/6 Mice

In the manuscript, the authors investigated the relative markers of colitis in the different acute, chronic and remission phases induced by dextran sodium sulphate (DSS) for understanding inflammatory conditions. In general, the authors have completed a reasonable study with very informative data on the relationships of inflammatory parameters and the observations of histopathological alterations at different stages of DSS colitis. The statistical analysis and graphic presentation also have been clearly showed the differences in details. These findings are very interested to us to understand the etiology of colitis, especially in the microflora. However, the findings of Lactobacillus species decreased with chronification of disease as indicated from the authors might be strengthened by taking a discussion and proposing the passible pathway to explain the cause and effect of Lactobacillus species numbers.

Specific comments:

1. The format of references should be revised in consistency, i.e. the ref. 1.

2. Line 215, -80 °C?

3. Fig. 3,
Please check the data of **** p < 0.0001 being determined in the figure.

Author Response

Point-to-point reply - Reviewer 2:

Reviewer: In the manuscript, the authors investigated the relative markers of colitis in the different acute, chronic and remission phases induced by dextran sodium sulphate (DSS) for understanding inflammatory conditions. In general, the authors have completed a reasonable study with very informative data on the relationships of inflammatory parameters and the observations of histopathological alterations at different stages of DSS colitis. The statistical analysis and graphic presentation also have been clearly showed the differences in details. These findings are very interested to us to understand the etiology of colitis, especially in the microflora.

Authors: We would like to thank the reviewer for the positive feedback on our work and hope to sufficiently implement all recommendations in the following.

Reviewer: However, the findings of Lactobacillus species decreased with chronification of disease as indicated from the authors might be strengthened by taking a discussion and proposing the passible pathway to explain the cause and effect of Lactobacillus species numbers.

Authors: We agree with the reviewer that this aspect should be strengthened in the discussion. Hence, we described the current understanding of the beneficial functions of Lactobacilli for the mucosal homeostasis in more detail and added a reference to an extensive review about this topic. We hope to thereby satisfy the reviewers recommendation.

Reviewer: The format of references should be revised in consistency, i.e. the ref. 1.

Authors: We fixed the text format for the bibliography and reference #1 in particular.

Reviewer: Line 215, -80 °C?

Authors: We added a minus sign in front of the temperature statement.

Reviewer: Fig. 3, please check the data of **** p < 0.0001 being determined in the figure.

Authors: Since there are indeed no statistically significant differences with p < 0.0001 in Figure 3, we deleted this p-value category from the figure legend. We also fixed similar issues in the figure legends for supplementary figures 2, 3 and 4.